# Assessment of Forest Ecosystem Services in Burabay National Park, Kazakhstan: A Case Study

**Sara Kitaibekova** [1,*], **Zhailau Toktassynov** [1], **Dani Sarsekova** [1], **Soleiman Mohammadi Limaei** [2] and **Elmira Zhilkibayeva** [3]

1  Department of Forest Resources and Forestry, Saken Seifullin Kazakh Agrotechnical University, Astana 010011, Kazakhstan
2  Department of Economics, Geography, Law and Tourism, Mid Sweden University, 851 70 Sundsvall, Sweden
3  Department of Forest Resources and Game Management, Kazakh National Agrarian Research University, Almaty 050010, Kazakhstan
*  Correspondence: saraorazbek@mail.ru

**Abstract:** The issues of forestry and the economic assessment of ecosystem services of forests, in the example of the national park "Burabay" in Kazakhstan, located in the northern part of the country, are investigated in this study. The relevance of forest ecosystem services, such as carbon fixation, oxygen emission, soil conservation from erosion and precipitation redistribution, are important environmental factors that contribute to the conservation of natural capital. Studies by domestic and international scientists show that the value of ecosystem services of forests is much higher than the cost of wood and its material products. Consequently, the ecosystem services of forests should be valued in monetary terms and considered in the context of the national wealth of the country. The main purpose of this research is to establish the value of non-market forest products while considering the prevailing natural and socio-economic conditions. The methods of the ecosystem approach for assessing the above ecosystem services of forests in value terms were implemented, and the real value of the multifunctional value of forests has been revealed. The results obtained can be used in planning measures to improve the sustainability of forests, ecotourism organizations and management decision-making.

**Keywords:** national park; forest ecosystem; pine forests; ecosystem services; ecotourism; soil protection; stock of plantings; carbon sequestration; oxygen emission; recreation

## 1. Introduction

Since the last decades of the twentieth century, the community has been especially concerned about the issues of disruption of the natural balance, loss of biological diversity, depletion of natural resources, global warming and other negative processes taking place around us in the world [1–3].

The purposeful activities of the United Nations (UN), numerous environmental organizations, scientists and practitioners, united under its leadership, have yielded positive results. Their first major achievement on a global scale in this area was the decisions of the Global Conference in Rio de Janeiro (1992) under the general and capacious name "sustainable development".

Among other countries, Kazakhstan has undertaken obligations to fulfill the requirements of fundamental interstate documents, such as the "Convention on Biological Diversity", "Convention to Combat Desertification" and "UN Framework Convention on Climate Change" [4] and others. Moreover, relevant legislative acts have been adopted to implement them.

The assessment of natural resources is a difficult economic task, and a lot of attention has been paid to it by scientists from many countries [5–8]. Subsequently, assessment methods were developed and experimental calculations were carried out on the economic

assessment of forest resources in a number of regions [9,10]. In Kazakhstan, Bayzakov conducted thorough scientific research in the field of cadastral and the economic assessment of forest resources [11,12]. In particular, he noted that the essence of most of the forest assessment works of the Soviet period proceeded from establishing the value of the forest as a source of wood, which almost did not affect its ecological and environmental properties. Therefore, in his work, he proposed to assess forests in the aggregate and divided them into two groups: wood resources and ecological properties.

In the economic evaluation of forests, it is still quite difficult to determine the significance of their ecological role, functions and useful properties, especially in terms of commercial value. In Kazakhstan, there is some experience in assessing natural resources in monetary terms. In particular, the natural resources of the Aksu-Zhabagly Nature Reserve in the South Kazakhstan region and the Altyn-Emel National Park in the Almaty region were evaluated (2005). Wood and non-wood forest products have more reliable cost estimates, but the ecosystem services of forests, including the so-called useful commercial properties and functions, are quite difficult to estimate.

In Kazakhstan, the potential possibilities of forests for carbon sequestration and oxygen emissions by living parts of woody plants in the territory of the Kazakh Upland were investigated by Boranbay [13]. In particular, Boranbay found that in terms of specific indicators of timber stock, phytomass and oxygen per hectare of forests in the Burabay National Park, they prevailed over specific average data for the entire Kazakh Upland. The works on carbon accumulation in artificial forest plantations in the green zone of the city of Astana were carried out by Tumenbayeva [14]. According to the results of their research, it was found that the plantations in question deposited carbon by USD 1134 per hectare.

The problem of increasing the recreational stability of pine plantations in the Shchuchinsk-Borovskoye resort zone, based on the use of a set of diagnostic indicators, was studied in detail by Dancheva. Because the main role of the studied plantations was to use them for recreational purposes, coupled with the peculiarities of the growth and development of these plantations associated with the growth in severe arid conditions, meant they classed them as a unique natural phenomenon.

The novelty of the results obtained by Dancheva and their scientific value lies in the fact that significant amendments have been made to the functional zoning of pine plantations of the Kazakh Upland [15]. The analysis of the state of pine plantation components depends on the degree of recreational impact in various forest site conditions and scientifically substantiated the proposed forestry management measures, which, when implemented, will reduce the negative impact of recreation on pine plantations and increase their recreational stability and attractiveness.

Scientists Mohammadi Limaei et al. from Iran, who studied the Saravan Forest Park, noted that non-market goods and services are difficult to evaluate and that there is no bargaining for them in the market. Such goods and services include recreational values, air quality, scenic values, environmental resources and services. There are several methods for assessing recreational value and ecosystem services [16].

Based on the results obtained from studies conducted to determine the economic value of tourism and recreation in a large network of protected areas, Australian scientists Heagneya et al. [17] found that recreational services provided by protected areas can be an order of magnitude, and possibly even more than the enabling uses that traditionally attributed economic value. To achieve social benefits, it is proposed to take this provision into account when distributing land use. When studying the recreational value of forest parks, the travel cost method is often used. In particular, Ghana scientists Ankomah, Emmanuel, Adu and Kofi Osei [18] and Iranian scientists Pirikiya, Amirnejad, Oladi and Ataie Solout [19] used this method in their studies. They believed that the results of this method were most acceptable when making managerial decisions on the conservation and sustainable management of natural resources, organizing additional services in parks, enhancing advertising and educating the public about the importance of recreation. Studying the question of the willingness to pay visitors to the Masaryk Forest Křtiny (TFE Křtiny) Educational Forestry

involved using the conditional valuation method as part of the TFE Křtiny recreational potential economic evaluation study [20]. Research results of the questionnaire showed that visitors were not very willing to pay for recreational features, especially since the forests in the TTI Krština area are perceived as public property, meaning access to them should be free.

The economic aspects of assessing forest functions, including the issues of forest use in the concept of sustainable development, were studied by scientists from Poland [21]. According to this study, there should be a balance between the needs of society, the requirements of the forest for sustainable growth, the appropriate technologies and practical forest management procedures.

The silvicultural and economic role of forests and adjacent territories should be considered as the provision of various ecosystem services. Significant scientific interest in terms of methodological approaches and problem-solving are the results of research on the assessment of ecosystem services of the forestry fund of the Karkaraly National Park [22] in the Karaganda region. Here, the ecosystem approach, or the classification of services and the identification of their volumes and cost estimates, which are generally accepted in international practice, is applied. A similar methodology was followed by scientists at the Kazakh National Agrarian University [23] when assessing ecosystem services on the example of the Bakanassky State Forestry in the Almaty region.

Previously, similar work was carried out in this region, but they mainly mentioned the study of the recreational capacity of forest plantations. For example, Boranbay's studies were more concerned with the assessment of the forests of the Kazakh Upland as a whole, while the object of Tumebayeva's research was artificial plantations. The purpose of this study was to complete a comprehensive economic assessment of ecosystem services of the Burabay National Park for the absorption of carbon dioxide and the emission of oxygen, the redistribution of precipitation and soil protection and the recreational potential in monetary terms. The forest activities in the Republic of Kazakhstan are financed from the state budget. The novelty of this research lies in the improvement of scientific research in Kazakhstan on forests and the economic assessment of resort forests, in connection with a significant increase in their ecological and recreational role.

## 2. Materials and Methods

*Study Area*

Burabay State National Natural Park is located in the north of Kazakhstan in the Akmola region, and its territory is located in the Kokshetau Upland, belonging to the Kazakh Upland (Figure 1). The type of climate in the national park "Burabay" is sharply continental. General indicators of continentality are a significant amplitude of winter and summer, day and night temperatures and a significant excess of evaporation over precipitation [24].

The total area in the Burabay National Park is 129,900 hectares, of which 79,300 hectares are covered with forests (Figure 1). The main forest-forming species are Scots pine (Pinus sylvestris) and birch (Petula pendula sp.), of which the proportion is 65.6% and 29.0%, respectively. There are small areas of aspen (Populus tremula) and other tree and shrub species. On the territory of the park, there are the following types of game animals: Cervidae, Capreolus, Alces, Lepus, Sciurus, Marmota, Ondatra zibethicus, Vulpes, Canis lupus, Tetrao urogallus, Lyrurus tetrix and Perdix perdix.

To systematize the areas of different categories of land and to establish functional zones, the materials from the latest forest inventory of the Burabay National Park were used [25]. General information is presented in Table 1.

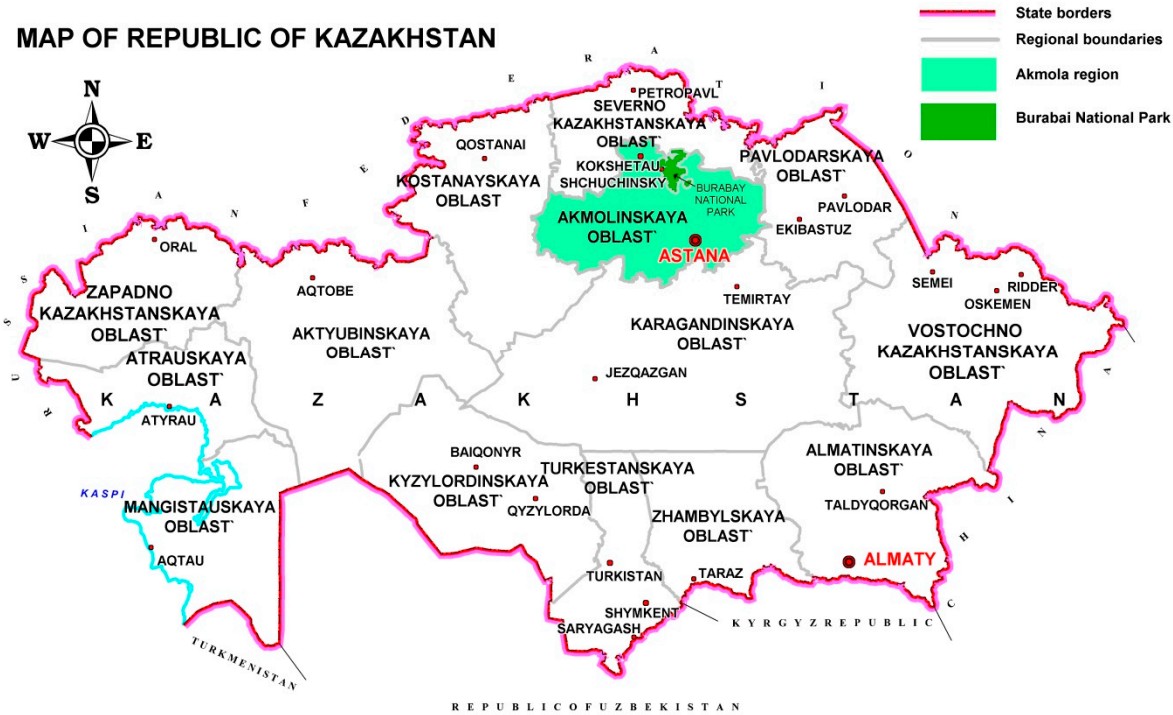

**Figure 1.** The map of the territory of the national park "Burabay".

**Table 1.** Summary table of the areas of land and timber reserves of the forest fund of the national park.

| Distribution of Forest Land by Category | Area (ha) | Specific Gravity, in % | Distribution of Forested Land by Dominant Species | | |
|---|---|---|---|---|---|
| | | | Species | Area, ha | Specific Gravity, in % |
| Forest land: | 91,218.6 | 70.2 | Pine | 51,987.3 | 65.6 |
| Including covered | 79,284.8 | 61.0 | Birch | 23,030.1 | 29.0 |
| Not covered | 11,933.8 | 9.2 | Aspen | 2915.9 | 3.7 |
| Non-forest land | 38,716.4 | 29.8 | Other tree species | 150.4 | 0.02 |
| Total area | 129,935.0 | 100.0 | Bushes | 1201.1 | 1.51 |
| Total stock | 13,044.6 th.m$^3$ | | Total | 79,284.8 | 100.0 |

From a variety of natural ecosystems, five types of services were selected for assessment: carbon fixation, oxygen evolution, soil protection from erosion, atmospheric precipitation redistribution and recreation (Table 2).

**Table 2.** The list of ecosystem services selected for economic assessment in the Burabay National Park.

| Ecosystem Service | Type of Ecosystem Services | Methods for Assessing Ecosystem Services |
|---|---|---|
| $CO_2$ sequestration | Regulating | International price of 1 ton of carbon [26] |
| Oxygen emission | Regulating | Industrial production cost [27] |
| Soil protection | Regulating | Restoration costs [28] |
| Redistribution of precipitation | Regulating | Market prices |
| Recreation | Cultural | Market prices |

To conduct an economic assessment of the selected ecosystem services from a variety of available methods, this study selectively used market prices, officially approved and current rates of payment, restoration costs, industrial production costs and internationally recognized prices.

To calculate the mass of the fixed carbon and oxygen released, this study used Komissarov's method [29], which is used to carry out calculations of the atomic weights of chemical elements in the composition of carbon dioxide ($CO_2$).

Considering the atomic weight (E) of carbon ($E_c$), 12, and oxygen ($E_o$), 16, the molecular weight (M) of carbon dioxide ($M_c$), 44, and oxygen ($M_o$), 32, it is easy to determine the carbon dioxide absorbed by plants and the emission of oxygen upon the assimilation of 1.0 tons of carbon. Denoting the amount of carbon dioxide using x and oxygen with y, the following proportion was compiled:

$$x = \frac{Mc}{Ec} \tag{1}$$

After substituting the numerical values of $M_c$ and $E_c$ in Equation (1), it was determined that x = 3.666 t.

A new ratio was then created: y: 3.666 t = 32:44, where y = 2.666 t, considering the photosynthetic coefficient being equal to 1.05 and the emission of oxygen into the atmosphere being calculated as $2.666 \times 1.05 = 2.799$ t. Additionally, this ratio considers that the carbon content in plants is about 50% and that forest plantations absorb 1.83 tons of carbon dioxide (3.666 tons × 0.5) per 1 ton of dry organic matter growth and release 1.40 tons of oxygen (2.666 tons × 1.05 × 0.5).

The recreational value of the forests was revealed by asking a group of visitors the following questions:

Question 1: Should Kazakhstan not pursue development programs that harm the environment, no matter how small the environmental cost is?

Question 2: Should we not invest in the environment and sacrifice our incomes and standard of living so that the next generation can benefit from the planet and the animals on it?

Question 3: Will the cost of the SNNP Burabay be the same, with or without animals?

Question 4: To what extent did the presence of animals in the park affect your visit to a specially protected natural area?

Question 5: Should citizens pay for SNNP Burabay and its nature reserves, even if they do not visit or use them?

Question 6: Do animals have a right to exist, even though they may be of no use to humanity?

## 3. Results

### 3.1. Carbon Fixation Value

To determine the value of forests for carbon fixation, the total stock (V) of wood pulp in the wooded area and the average density coefficient of wood in the air-dry state (D) were used (Equation (2)):

$$V \cdot D = 13{,}044 \cdot 0.51 \tag{2}$$

The total stock of wood pulp in the wooded area (V) was equal to 13,044.6 $m^3$. The average density coefficient of wood in the air-dry state was 0.51 [30].

By substituting the numerical values of V and D in Equation (2), the result was 6652.7 tons.

Considering that the carbon content in wood is about 50% [31], it was found that the mass of fixed carbon was $6652.7 \times 2 = 326.4$ tons (Table 3). The price for 1 ton of carbon was considered USD 69.33 [26]. In the same way, the monetary value of sequestered carbon in Retesat National Park, Romania was calculated [32].

**Table 3.** Determination of the value of the forests of the Burabay National Park for carbon fixation.

| Total Stock in the Forested Area, Tons | Fixed Carbon Mass, Tons | Price for 1 Ton of Carbon, USD | The Total Carbon Fixation Value of Forests, USD |
|---|---|---|---|
| 6652.7 | 3326.4 | 69.33 | 230,619.3 |

The results of the calculations of the value of fixed C according to the above methodology are shown in Table 3.

Consequently, the total value of the forest in terms of carbon fixation in the amount of 3326.4 tons was estimated at USD 230 619.3. In terms of 1 ha of forest-covered area, the mass of fixed C was 41.95 t/ha, and the value of C fixation was estimated at USD 2900.

### 3.2. Calculation of Oxygen Release

To calculate the total increase of dry organic matter, Equation (2) was used. The total stock of plantations in the wooded area was equal to 13,044.6 $m^3$ multiplied by the average coefficient of the specific gravity of wood in a dry state (0.51) and it was equal to 6652.7 tons.

The result was multiplied by 1.40, meaning that during the accumulation of this mass, forest plantations released 9313.8 tons of $O_2$. With the price of industrial production of one ton of oxygen at KZT 110,000 [27], the total estimate of the indicated volume of $O_2$ was equal to USD 2382.6. The oxygen-producing function of forests was estimated at USD 30,000 per hectare of forested area.

### 3.3. Value of Soil Protection

The value of the forests of the Burabay National Park for soil protection was then calculated. These services were in the regulatory category and were specific to forested land (79,285 ha), and the costs of reforestation were used to estimate them as, without them, soil erosion can occur. Moreover, it has been proven that the thicker the forest, the better it strengthens the soil, resulting in a weaker development of the erosion process.

The plantation costs for regeneration per hectare of forest trees (pine and birch), according to the working draft developed by the Kazgiproleskhoz Design Institute [24] was KZT 322,000. When developing working drafts of forest plantations, in accordance with current requirements, the soil conditions were studied in detail, including an analysis of samples of their forest suitability and the selection of an appropriate assortment of woody and shrubby plants for planting. Consequently, when the costs (C) were spread over the entire forested area, their total amount was USD 59,371.5 (79,285 ha·KZT 322,000: KZT 430). Moreover, if this amount was divided by the area of forested land (A), then using the following equation:

$$M = \frac{C}{A} \qquad (3)$$

the average conservation value (M) per hectare of forests would be USD 748.68.

### 3.4. Redistribution of Precipitation

The redistribution of precipitation, including its transfer to groundwater and simultaneous purification, is one of the important services of forests. The volume of this service (L) can be calculated based on the fact that about 10% (37.9 mm) of all precipitation (P) during the year penetrated the soil [33], while also being purified at the same time, replenishing groundwater with an additional portion of water that can be calculated using the following relation:

$$L = P \bullet A \qquad (4)$$

By substituting the numerical values of P and A in Equation (4) the results would be 30.0 million $m^3$.

Such fresh water in the Shchuchinsk district, depending on the source of supply, varies between KZT 107 and KZT 194 per $m^3$. For our calculations, the average cost of KZT

150 per m$^3$ of clean water was used. Consequently, the services of a forest area equal to 79,285 hectares for the redistribution of precipitation can be estimated at USD 10,465.1 (30.0 million m$^3$ *x* KZT 150: KZT 430) or USD 132 per hectare of forest.

### 3.5. Recreational Value

The combination of mountainous terrain, forests, lakes and other rich natural diversity has formed a unique landscape, which is attractive for recreation and recreational tourism. During a relatively long period of recreational use of the territory, the existing infrastructure of sanatorium and hotel complexes and a road network has developed here, which successfully provides services to vacationers and tourists.

However, an uncontrolled increase in the number of visitors can exceed the recreational capacity of the territory and have a negative impact on the environment, primarily on the forest ecosystem. Meanwhile, there is a tendency to increase the flow of tourists, at the expense of the population of the rapidly developing capital of Astana, from other parts of Kazakhstan and neighboring Russia. The dynamics of the changes in the number of visitors to the national park over the past decade are shown in Figure 2. In the period from 2010 to 2019, the number of visitors increased by 495,600 people, or fourfold, after a slight decrease during some years. The sharp decline in visitors in 2020 was due to the pandemic restrictions, but we expect the number to rise again soon. In light of this increase, there is a need for constant monitoring of the state of forest plantations and the assessment of their potential for recreational services.

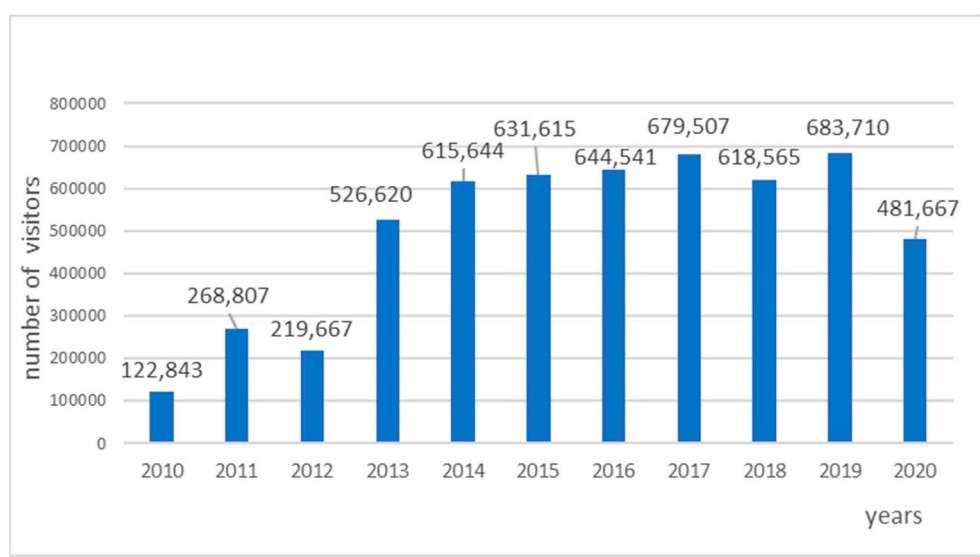

**Figure 2.** Number of visitors to the national park "Burabay". (Source: Department of Statistics for the Akmola region).

To solve these problems, the distribution of the territory of the national park was studied and split into functional zones. Four zones were established:

- The reserve regime (14,069.6 ha—10.8%) included the best preserved and unique areas of nature with indigenous vegetation and habitats of rare and endemic species of flora and fauna;
- Ecological stabilization (43,009 ha—33.1%) included stable, typical non-transformed areas or those slightly disturbed by economic activity ecosystems and natural objects, including habitats of rare and endemic species of flora and fauna, as well as areas of water and lakes;
- Tourist and recreational activities (12,110.2 hectares—9.3%) included areas confined to places of traditional rest and recreation. It also included motorways, lake shores and unique and attractive locations;

- Limited economic activity (60,746.2 hectares—46.8%) included mainly inter-forest spaces, locations of cordons, areas bordering on outside land users and settlements, pasture and hayfields, as well as split forests located in the park's buffer zone.

Based on the available materials, it was also established that there were twenty-four approved tourist routes in the national park, of which seventeen were hiking routes, four were bus routes, one was a bicycle route, one was a horse route and one was a water route (data of the National Park, 2020).

The assessment of non-market services of the Saravan National Park in Iran, carried out by Mohammadi Limaei et al. [16], applied the willingness to pay method, which involved conducting a survey and asking visitors a wide range of questions, including how much they were willing to pay for specific environmental services. In addition, our research had similar goals. In some cases, people were asked the amount of compensation they would be willing to accept for refusing specific environmental services, and the "declared preference" was also considered since people were asked to declare their willingness to pay based on a specific hypothetical scenario and the description of environmental services. The necessary information was obtained from the respondents during the survey, including product descriptions and details about the demographic characteristics of the respondent. The answers to the questions in the questionnaire are given further in the text.

To assess the Burabay National Park, in the summer of 2020, 950 respondents were interviewed, including 81% male respondents and 19% female respondents. These respondents were placed in the following employment categories (Figure 3):

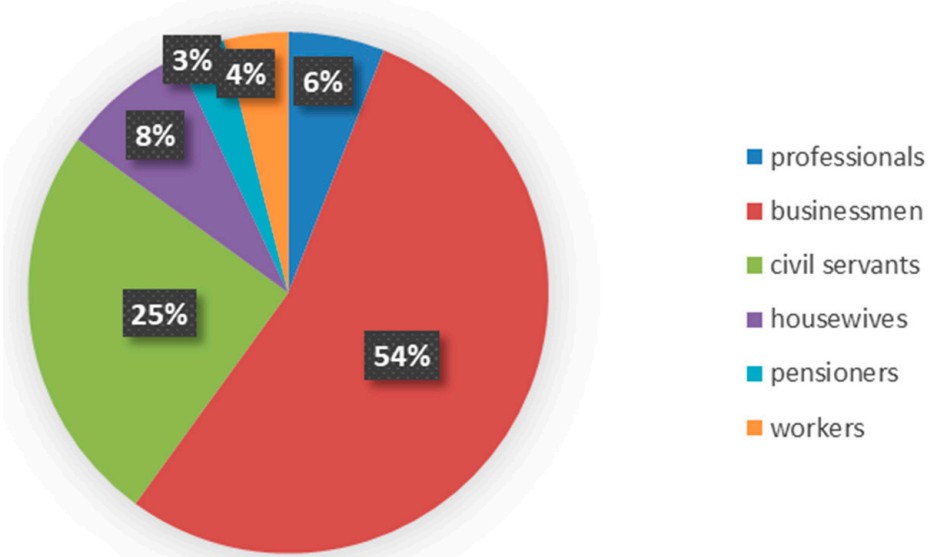

**Figure 3.** Percentage of respondents by category.

In relation to the environment and natural resources, the respondents were asked six questions each and the following answers were received:

In response to Question 1, 40.4% of respondents believed that Kazakhstan should not pursue development programs that were detrimental to the environment, no matter how small the environmental costs were. Additionally, 15% were undecided on this issue, but the majority of respondents (44.6%) were in favor of the development of economic programs using natural resources, without which it would be impossible to ensure the material well-being of society and improve their standard of living (Figure 4).

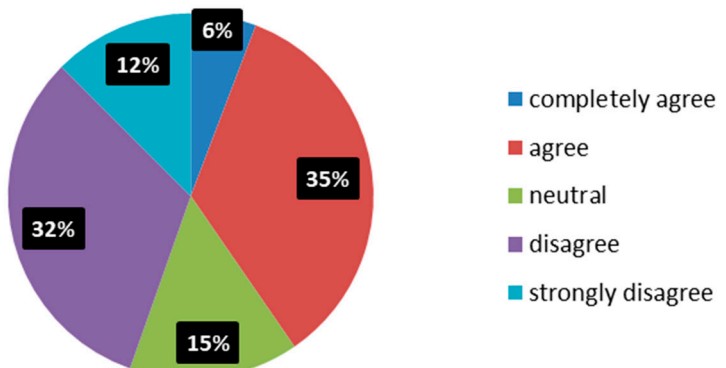

**Figure 4.** Respondents' answers to the question about the implementation of development programs.

Concerning Question 2, only 18.3% of the total number of respondents were unwilling to invest in environmental conservation, while another 10.2% provided neutral responses. Additionally, the majority of respondents (71.5%) were in favor of investing in the environment, since they would use it to benefit themselves and their descendants (Figure 5).

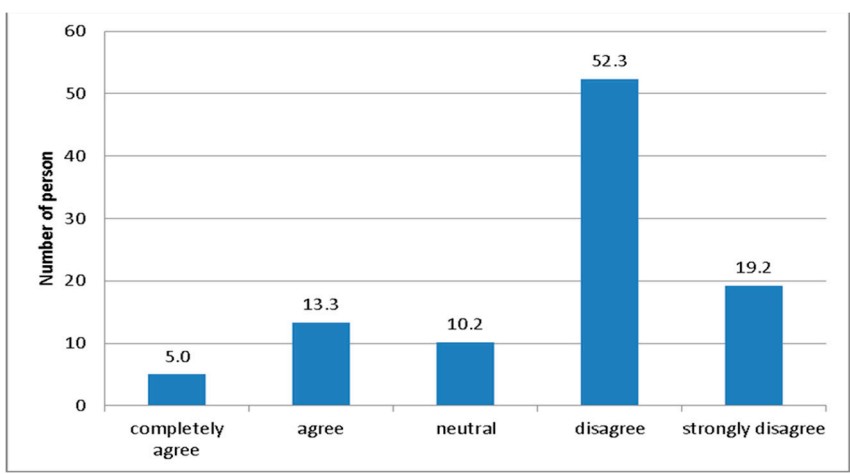

**Figure 5.** Respondents' answers regarding whether they would like to invest in the environment.

When answering Question 3, most of the 950 questioned respondents (66.4% of them) were confident that the value of the national park with wild animals and birds would be much higher than without them (Figure 6). This is due to the fact that without wild animals, the uniqueness and natural harmony of the park would be lost, which would naturally lead to its devaluation.

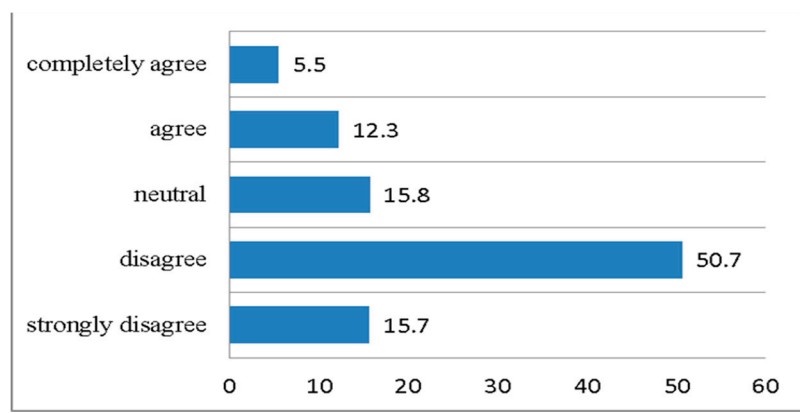

**Figure 6.** Will the value of Burabay National Park be the same, with or without animals?

A significant number of the respondents (72%) who answered Question 4 visited to see not only forests, lakes and mountains, but also wild animals, birds and other mammals during recreational and hiking trips. Many of them said that they had heard quite a lot about the Burabay National Park and had seen videos and pictures, but they had a great desire to see its beauty with their own eyes, breathe in the healing forest air and swim in the beautiful lakes (Figure 7).

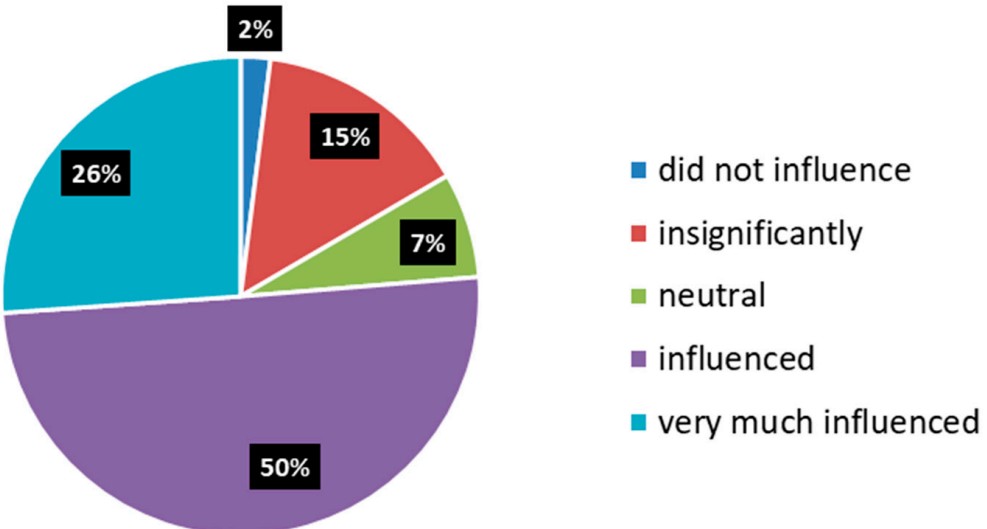

**Figure 7.** The degree of influence of the presence of animals on arrival.

It followed from the answers to Question 5 that the majority of people (59.5%) were sympathetic about the need to develop specially protected natural areas, which is a key part of biodiversity conservation, and that these areas have, and in the future will have, an indirect beneficial effect, even on those who do not yet understand its general usefulness (Figure 8).

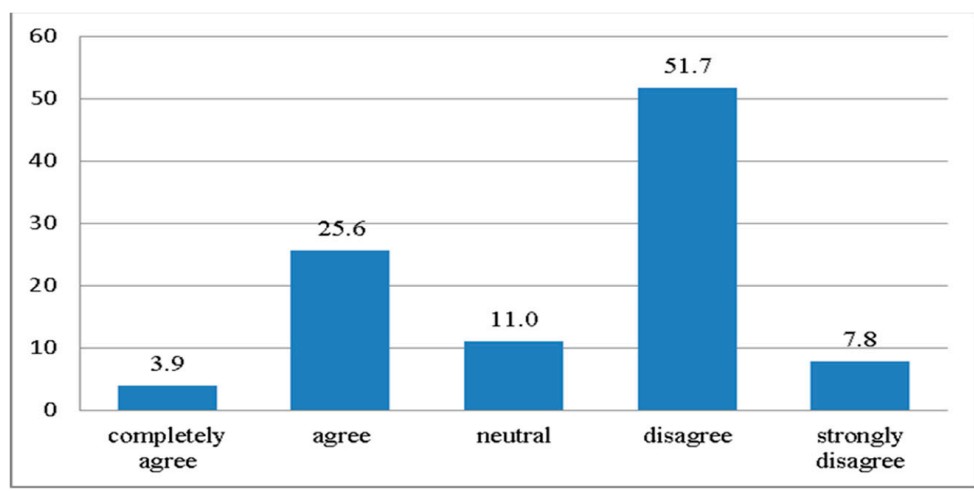

**Figure 8.** Respondents' answers to the question on the paid maintenance of specially protected natural areas.

Animals are an important part of the natural complex. They are always useful both to nature and to humanity since, without them, the natural balance will be disturbed, which can lead to irreversible processes. The majority of the surveyed respondents (85.6%) answered the sixth question by stating that animals have the right not only to exist but also to protect and reproduce (Figure 9).

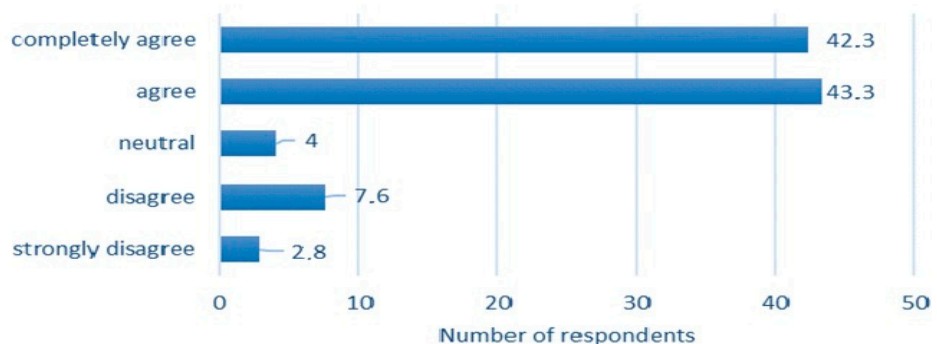

**Figure 9.** Answers of respondents to the question about the right of animals to exist.

## 4. Discussion

Studies of ecosystem services of forests in Kazakhstan show that its results, obtained in value terms, become a real expression of the value of natural capital and create the basis for its accounting in the composition of the national wealth of the country.

The conducted studies are the continuation of scientific works in our republic and abroad in this direction. Their results allowed us to more reliably determine the value of ecosystem services and provided a basis for improving their quality and expanding the range of services.

This statement agrees with the opinions of Iranian scientists who conducted similar research in the forest park of Gale Rudhan [34]. The majority of respondents in Kazakhstan are sympathetic to the fact that it is necessary to pay for the development of specially protected natural areas (59.3%). The same opinion is held by 62% of visitors to the forest park "Hermia" of Kosovo [35]. To increase the satisfaction of the visitors of the forest park "Akasawa", Japanese scientists recommended considering the number of visits (experience) and their duration when planning hiking routes [36]. Romanian scientists made another valuable suggestion; they noted that financial efforts to manage the ecosystems these services present to us can be supported by implementing financial mechanisms aimed at directing the value of ecosystem services to the management of these ecosystems [32].

According to our data, the mass of fixed carbon per 1 hectare of the forested area was 41.95 t/ha. The same amount (42.06 t/ha) was determined by scientists in Vietnam but in the poor and restored forests in Ba Be National Park [37].

Summarizing the results obtained, it can be argued that they match with the results of scientists from Iran, Poland and other countries. In line with current research, more people were interested in financially supporting the Burabay National Park. The research results showed that the citizens of Kazakhstan were very interested in the development and protection of specially protected areas, such as the Burabay National Park. Additionally, a significant number of the respondents who answered the questionnaire were concerned about the situation and asked for increased attention from the decision-making bodies regarding the future development of the park as a nationwide and valuable asset. The total value of the regulatory ecosystem services reviewed was USD 302.8 million.

## 5. Conclusions

The authorized state bodies of Kazakhstan on the environmental safety of the country are actively looking for new ways to combat global warming and desertification and preserve biodiversity. In this regard, there is considerable interest in scientific developments to establish the value of natural objects and assess their ecosystem services in value terms. The results of this study can be used in the following ways:

- Profile ministries, as well as the Committee for Forestry and Wildlife Protection of the Republic of Kazakhstan and its subordinate organizations, can develop standards aimed at the prudent use of natural resources to improve the state of the unique forests of the resort zone, as well as solve land use and other economic issues;

- Based on the results of this research, it is possible to determine the potential of the region in the global balance of production and consumption of resources such as carbon, as well as the possibility of buying or selling these resources in the future, based on the intra-republican and interstate quotas established for them;
- The experience of advanced countries shows that significant material and financial costs for improving the infrastructure of the national park are gradually paid off, primarily due to an increase in the flow of tourists and vacationers, not only in the summer but also in the winter. It is known that preserved ecosystems are a genetic reserve of living capital, which also perform an invaluable role in the circulation of substances in nature and provide a variety of ecosystem services. The results of the assessments of these services, carried out by scientists from Kazakhstan and other countries, including the US, show that they are much more expensive than the costs of their preservation;
- The further development of the tourism industry will be important, ensuring an increase in the use of tourism services, which includes tours, tourist and excursion services and goods. A significant part of the income from these tourism services will settle in the region and have a beneficial effect on its further development. There is another benefit of this process, which involves providing employment for local residents and increasing their personal income;
- A trend has been established that the number of vacationers and tourists will increase annually, including international tourists, increasing the recreational load on forests. This means it is necessary to develop a set of forestry measures to increase their recreational capacity by improving tourist routes and implementing forest conservation and firefighting measures;
- It is necessary to increase the attractiveness of routes by improving their infrastructure, enhancing the service culture, creating special forest plantations to close unsightly places and improving the food supply for wild animals and birds to increase their numbers and species.
- This article was written based on the research results of a scientific project for grant funding of the Ministry of Education and Science of the Republic of Kazakhstan for 2018-2020, No AP05134807 "Landscape and environmental assessment of the state of green spaces in the city of Astana and suburban areas, ways to optimize the landscaping system".

**Author Contributions:** Conceptualization, S.K., Z.T., D.S. and S.M.L.; data collection, S.K., Z.T., E.Z.; methodology, Z.T.; formal analysis, Z.T., S.M.L.; writing draft S.K. and Z.T., writing review and editing S.M.L.; supervision; E.Z. This study results from the independent analytical research of the authors' organizations and institutions (p.1). All authors have read and agreed to the published version of the manuscript.

**Funding:** This research received no external funding.

**Institutional Review Board Statement:** Not applicable.

**Informed Consent Statement:** Not applicable.

**Data Availability Statement:** The data are available on request from the corresponding author.

**Acknowledgments:** The investigations at the national park "Burabay" were supported by the Saken Seifullin Kazakh Agrotechnical University.

**Conflicts of Interest:** The authors declare no conflict of interest.

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
