# Peer review of "Assessment of Forest Ecosystem Services in Burabay National Park, Kazakhstan: A Case Study"

_sustainability, doi:10.3390/su15054123_

Round 1

Reviewer 1 Report

Title: Assessment of forest ecosystem services (Case study: national park of Burabay, Kazakhstan)

General comments: This is an interesting topic for investigation. The methods are not very clear. There is a need to be more organized, methods and results cannot be mixed. The conclusion is quite vague and mostly on future recommendations.

Specific comments:

Abstract: 

Sentence of Line 22 should be first sentence

Abstract to be better organized. Information on methods, detailed results, discussion and conclusion should be given. There is tendency to repeat the objectives several times.

Introduction

line 43, 'global' rather than 'world'

line 46, need references to support

lines 47 to 51 more recent literature needed, there are more recent developments and updates

line 56, need references to support statements

lines 69-71, statement not clear

line 78. too general statement

line 90, reference style not clear, sometimes it is by numbers, sometimes names of authors

line 95 not clear 'whose results'

line 97, sentence structure 'it was analyzed' not clear

line 105, avoid use pf 'etc'

line 109, line 115 references and style not clear

line 115-126, it is not clear whether authors are evaluating methods

page 4 the introduction is way too long

materials and methods:

lines 166-170, general names of animals are given. Scientific names should be included in general.

Fig 1: to check if this is according to Journal specifications, font size is small

Table 1: main indicators of what, it is not clear;

last line check numbering, decimal places to be reported as as dot and not as comma

Table 2:  The information should be supported by references

line 197, avoid use of 'we'. why is it in future tense

lines 204 to 214, there should be alternate ways of reporting, use of formula, it is prose like

Table 3 use of decimal places rather than comma

line 251, sentence structure

lines 257- 280 alternative ways of reporting needed, too much prose like

Page 9, figure 2, sources of values to be reported

line 323 need to use specific references

line 327, year of reference to be included

line 330 details of survey questionnaire design needed

Page 10, questions and answers included, questions should be in methods, may be a copy of the questionnaire to be included as supplementary material

Figure 4 pie charts should be flat; some data processing needed.

Line 362, question to be checked, avoid use of apostrophe

figure 5, typo, 'disagree'

Figure 6, to include number of samples

Figure 7, avoid use of apostrophe

line 422, question to be checked...'useless to humanity'

line 428 sentence structure to be checked

Discussion

line 436 too general statement

line 440 check sentence structure

line 447 are results being presented here, it is not clear

line 490-495, is it a recommendation

conclusion

Needs to be more concise

line 525, do authors mean 'value,

The statements appear mostly as recommendations

references

Need more uptodate ones to support statements.

formatting to be checked, e.g. lines 588, lines 592

referencing of webpages to be checked

line 613, why 'ORCID@ included.

Author Response

Thank you for your comments! Yes, we agree that we need extensive editing of English language and style. We’ll use language editing service of “Sustainability” journal

Abstract: Sentence of Line 22 should be first sentence – changed as the first sentence

Introduction

line 43, 'global' rather than 'world' – changed into “global”

line 46, need references to support – added literature [1,2,3]

lines 47 to 51 more recent literature needed, there are more recent developments and updates – added recent literature [4]

line 56, need references to support statements – added [4]

lines 69-71, statement not clear - In particular, he noted that the essence of most forest-evaluation works of the Soviet period proceeded from the establishment of the value of the forest as a source of timber and almost did not touch its ecological and environment-forming properties. Therefore, in his works, he proposed evaluating forests in an aggregated manner, dividing them into two groups: wood resources and ecological properties.

line 78. too general statement - In the general chain of economic evaluation assessment of forests, it is still quite difficult to determine the significance value of their ecological role, functions and useful properties, especially in terms of value.

line 90, reference style not clear, sometimes it is by numbers, sometimes names of authors – it is changed

line 95 not clear 'whose results' – Dancheva’s results

line 97, sentence structure 'it was analyzed' not clear - The analysis of It was analyzed the state of the components of pine plantations components depending on the degree of recreational impact in various forest site conditions and scientifically substantiated the proposed forestry management measures activities, the implementation of which will reduce the negative impact of recreation on pine plantations and increase their recreational stability and attractiveness [7].

line 105, avoid use pf 'etc' - deleted

line 109, line 115 references and style not clear – changed

line 115-126, it is not clear whether authors are evaluating methods – yes, we are evaluating methods

page 4 the introduction is way too long- we shortened a little bit

Materials and methods: lines 166-170, general names of animals are given. Scientific names should be included in general – changed into scientific names - Cervidae, Capreolus, Alces, Lepus, Sciurus, Marmota, Ondatra zibethicus, Vulpes, Canis lupus, Tetrao urogallus, Lyrurus tetrix, Perdix perdix

Fig 1: to check if this is according to Journal specifications, font size is small – we increased the size of letters in the map on page 6

Table 1: main indicators of what, it is not clear - Summary table of areas of land and timber reserves of the national park's forest fund last line check numbering, decimal places to be reported as dot and not as comma – changed (91 218.6) Table 2: The information should be supported by references – included additional references [26,27,28]

line 197, avoid use of 'we'. why is it in future tense - changed into "it was selectively used marked prices'

lines 204 to 214, there should be alternate ways of reporting, use of formula, it is prose like - Considering the atomic weight (E) of carbon (Ec )-12 and oxygen (Eo) -16, the molecular weight (m) of carbon dioxide (mc) -44 and oxygen (mo) – 32, it is easy to determine the carbon dioxide absorbed by plants and emission of oxygen upon assimilation of 1.0 tons of carbon. Denoting the amount of carbon dioxide through - x, and oxygen through - y, the proportion is compiled: x: 1.0 = x=mc/Ec (1) If we substitute the numerical values of mc and Ec in Eq. (1), the results will be x = 3.666 t. Now we create a new ratio: y: 3.666 t = 32:44, whence y = 2.666 t, considering the photosynthetic coefficient equal to 1.05, emission of oxygen into the atmosphere made up (2.666x1.05= 2.799 t.). Knowing that the carbon content in plants is about 50%, forest plantations absorb 1.83 tons of carbon dioxide (3.666 tons x 0.5) per 1 ton of dry organic matter growth and release 1.40 tons of oxygen (2.666 tons x 1.05 x 0.5), whence y = 2.666 t, considering the photosynthetic coefficient equal to 1.05, emission of oxygen into the atmosphere made up (2.666x1.05= 2.799 t.) . Knowing that the carbon content in plants is about 50%, forest plantations absorb 1.83 tons of carbon dioxide (3.666 tons x 0.5) per 1 ton of dry organic matter growth and release 1.40 tons of oxygen (2.666 tons x 1.05 x 0.5).

Table 3 use of decimal places rather than comma - changed

line 251, sentence structure - The results of calculations of the value of fixed C according to the above methodology are shown in Table 3.

lines 257- 280 alternative ways of reporting needed, too much prose like - To calculate the total increase of dry organic matter, Eq.2 was used. The total stock of plantations in the wooded area was equal to 13.044.6 thousand m3 multiplied by the average coefficient of the specific gravity of wood in a dry state (0.51) and it was equal to 6.652.7 thousand tons. The result is multiplied by 1.40, i.e. during the accumulation of this mass, forest plantations released 9 313.8 thousand tons of О2. With the price of industrial production of one ton of О2 at 110 thousand tenge [27], the total estimate of the indicated volume of О2 will be equal to 2.382,6 thousand US dollars (considering the ex-change rate, 9.313.8 thousand tons x 110 thousand tenge :430 tenge). The oxygen-producing function of forests is estimated to be at 30.0 thousand US dollars per hectare of forested area.

Page 9, figure 2, sources of values to be reported - it is shown number of visitors by years

line 323 need to use specific references - data of the National Park "Burabay" (https:parkburabay.akmol.kz)

line 327, year of reference to be included - 2020

line 330 details of survey questionnaire design needed- we changed

Page 10, questions and answers included, questions should be in methods, may be a copy of the questionnaire to be included as supplementary material - questions were moved to the methods

Figure 4 pie charts should be flat; some data processing needed-we changed into flat one

Line 362, question to be checked, avoid use of apostrophe figure 5, typo, 'disagree' - it was changed 

Figure 6, to include number of samples - 950 respondents

Figure 7, avoid use of apostrophe- it was changed

line 422, question to be checked...'useless to humanity'- Animals are an important part of the natural complex, they are always useful: both to nature and to humanity, since without them the natural balance will be disturbed, which can lead to irreversible processes. The majority of surveyed respondents (85.6%) answered to the 6th question with understanding as animals have the right not only exist but also to protect and reproduce. 

line 428 sentence structure to be checked - Respondents' answers to the question of paid maintenance of specially protected natural areas

Discussion

line 436 too general statement - Studies of ecosystem services of forests in Kazakhstan shows that its results, obtained in value terms, become a real expression of the value of natural capital and create the basis for its accounting in the composition of the national wealth of the country.

line 440 check sentence structure - The conducted studies are continuation of scientific works in our republic and abroad in this direction. Their results allowed to more reliably determine the value of ecosystem services and provided a basis for improving their quality and expanding the range of services. 

line 447 are results being presented here, it is not clear- The majority of respondents in Kazakhstan are sympathetic to the fact that it is necessary to pay for the development of specially protected areas (59.3%)

line 490-495, is it a recommendation- yes, they are reccomendations

Conclusion

Needs to be more concise - yes, we agree with you and we shortened

line 525, do authors mean 'value, - yes, "value"

The statements appear mostly as recommendations references Need more uptodate ones to support statements. formatting to be checked, e.g. - The results of assessments of these services, carried out by scientists from Kazakhstan and other countries, including us, showed that they are much more expensive than the costs of their reservation.

lines 588, lines 592 referencing of webpages to be checked- - we checked

line 613, why 'ORCID@ included - it was deleted

Reviewer 2 Report

Sustainability-2074540 submitted by Kitaibekova et al- “Assessment of forest ecosystem services (Case study: national park of Burabay, Kazakhstan)” investigated the issues of forestry and economic assessment of ecosystem services of forests of the national park "Burabay" in Kazakhstan. The authors concluded that the results obtained can be used in planning measures to improve the sustainability of forests, ecotourism organizations and making management decisions. The article has scientific and economic impact and emphasis on the novelty of this study. The whole writing quality of manuscript is well. However, some major revision suggestions are listed below:

Some sentences sound repetitive or wordy. Also, some paragraph of the materials and methods, and discussion are hard to follow/understand. I recommend a detailed re-reading to rephrase those sentences.

Results

·         The results section needs to be shortened with emphasis on important and economical findings.

·         I suggest omitting all sentences with is no significant effect observed.

Discussion

·         The section is hard to follow, I highly recommend restructuring it. I would suggest making every effort to explain more the results and not compare them with other studies.

·         In general, the production data are interesting but did not discussed well.

·         The discussion should include further arguments between studies or results found in the literature.

Conclusion

·         In general, the conclusion is little bit long. The authors should be more clear and direct.

References

·         References must be numbered in order of appearance in the text and follow the recommended journal Style.

·         There is no references reflecting 2022, it should be for references updating.

Author Response

Thank you for your comments!

Some sentences sound repetitive or wordy. Also, some paragraph of the materials and methods, and discussion are hard to follow/understand. I recommend a detailed re-reading to rephrase those sentences. - We agree with you and we have changed some of them. They are marked in yellow in the text.

Results

The results section needs to be shortened with emphasis on important and economical findings - we have shortened this section

 I suggest omitting all sentences with is no significant effect observed - we have omitted all sentences with "no significance effect observed". We changed some sentences and paid attention to this comment.

Discussion

The section is hard to follow, I highly recommend restructuring it. I would suggest making every effort to explain more the results and not compare them with other studies. - we structured some sentences and they are marked in yellow in the text

In general, the production data are interesting but did not discussed well.- Thank you! We paid pay attention to your comments

Reviewer 3 Report

Reviewer Blind Comments to Author:

This research paper (Sustainability-2074540-peer-review-v1) is a case study of forest ecosystem services that investigated the constrains of the national park of Burabay, Kazakhstan. This paper particularized the important environmental factors for the development of forest ecosystem services. It is one of the forerunner contributions to understanding the sustainability of forests and monetary terms of the national wealth of the country, which is most suitable for publication in Sustainability.

However, there are certain criticisms observed in this article that need further modification.

Abstract: There are significant lack of clarity with respect to the scientific objective of the present study. It is not properly defined in this section. This has to be re-written before publication.

Some issues need to be clarified as follows:

1.  Write the significance of the monetary policy on the conservation of the forest ecosystem adopted in Kazakhstan?

2.  What kind of opinion was received from the people for Burabay National Park?

3.  Some citations and references are incomplete.

4.  Some paragraphs are repeated in the article. Remove those sentences (see annotated pdf file).

Research methodologies and their parameters are systematically organized.

There are numerous grammatical corrections and the re-organization of sentences is commented on in the main article in the form of annotation. That all points should be considered before submission of your revised version of the paper.

I suggested this manuscript should be appropriate to recommend for publication in the Sustainability with minor revision.

Author Response

Dear Reviewer, sorry, I didn't notice your first comments which you attached. I changed all and marked in the text in green! 

Reviewer 4 Report

Review for “Assessment of forest ecosystem services (Case study: national park of Burabay, Kazakhstan)” by Sara Kitaibekova et al.

1.     Acronyms should be defined before use in the manuscript, e.g., “the UN” on Page 2, I believe it should be “the United Nations (UN)”.

2.     There is an underscore on Page 3 under “[8]” that should be removed.

3.     Please make sure the styles of citations in the manuscript are consistent. I see some of the citations are reference numbers (e.g., [1]), but some of them are authors and year (e.g., Heagneya, Rosec, Ardeshirid, Kovaca (2019)). For the latter, I suggest using the last name of the first author and the reference number, i.e., Heagneya et al. [2019]

4.     The map in Fig. 1 is very blurry and the font sizes in the maps are too small to be readable. Please improve the quality of the figure.

5.     Please avoid using some non-English letters in the manuscript is it is not necessary, e.g., “тыс” in Table 1.

6.     Letter “x” has been used many times in the manuscript to represent multiplication sign, which could be confusing to readers. To avoid confusion, please use multiplication sign instead.

7.     The style of Table 1 and Table 2 are very different (one with background color but the other without). I would suggest keeping them the same style, if there is not particular purposes for doing this.

8.     There should be a comma sign after “i.e.”, which is “i.e.,”. For example, on Page 7 “i.e. during the accumulation” should be “i.e., during the accumulation”.

Author Response

Thank you for your comments! They are very useful!

  1. Acronyms should be defined before use in the manuscript, e.g., “the UN” on Page 2, I believe it should be “the United Nations (UN)”. - Yes, it means "the United Nations"
  2. There is an underscore on Page 3 under “[8]” that should be removed.- It was removed
  3. Please make sure the styles of citations in the manuscript are consistent. I see some of the citations are reference numbers (e.g., [1]), but some of them are authors and year (e.g., Heagneya, Rosec, Ardeshirid, Kovaca (2019)). For the latter, I suggest using the last name of the first author and the reference number, i.e., Heagneya et al. [2019] - it was corrected
  4. The map in Fig. 1 is very blurry and the font sizes in the maps are too small to be readable. Please improve the quality of the figure. - The map was changed with a more readible size of letters.
  5. Please avoid using some non-English letters in the manuscript is it is not necessary, e.g., “тыс” in Table 1. - it was changed
  6. Letter “x” has been used many times in the manuscript to represent multiplication sign, which could be confusing to readers. To avoid confusion, please use multiplication sign instead. - we used x as multiplication sign
  7. The style of Table 1 and Table 2 are very different (one with background color but the other without). I would suggest keeping them the same style, if there is not particular purposes for doing this.- table was changed 
  8. There should be a comma sign after “i.e.”, which is “i.e.,”. For example, on Page 7 “i.e. during the accumulation” should be “i.e., during the accumulation”.- yes, we paid attention to this comment

Reviewer 5 Report

Dear Authors,

The manuscript need an extensive improvement. It seems more like a school report than a scientific report.

1- The Abstract needs to be more specific and informatic.

2- The methodology and presenting the data needs huge improvement. 

3- Conclusion needs major revision.

Please find my comments in attached file.

Regards,

Author Response

Thank you very much for your comments!

1- The Abstract needs to be more specific and informatic. - It was changed a little bit.

2- The methodology and presenting the data needs huge improvement. - We made corrections and they are marked with yellow colour. 

3- Conclusion needs major revision. - It was changed a little bit.

Please find my comments in attached file. - We made some coorections according to the marked words and sentences

Reviewer 6 Report

The manuscript entitled " Assessment of forest ecosystem services (Case study: national park of Burabay, Kazakhstan)" concerns the ecosystem services providing the National Park. The subjects of the manuscript fall within the scope of the Journal and provide knowledge for the scientific community. However, the paper shows several gaps and weaknesses that need addressing. Below are some comments:

The title of the manuscript is needed to change.

The abstract is also not written for the international audience of the Journal. It needs rewrites. 

The manuscript introduction needs to improve, and the reference cited in this section are not computing as per the Journal pattern.

The materials and methods section also needs to improve

The methods of estimating the value of the services written in the results section need to shift in the material and methods section.

The results of the manuscript are good.

The discussion of the manuscript is needed to improve by adding new references to benefit international readers of the Journals.

Author Response

Thank you very much for your comments which are very useful!

Below are some comments:

The title of the manuscript is needed to change. - the title was changed a little bit.

The abstract is also not written for the international audience of the Journal. It needs rewrites. - it was changed a little

The manuscript introduction needs to improve, and the reference cited in this section are not computing as per the Journal pattern.- introduction was changed a little and reference put 

The materials and methods section also needs to improve - it was improved and marked in yellow colour

The methods of estimating the value of the services written in the results section need to shift in the material and methods section. - it was changed

The results of the manuscript are good.

The discussion of the manuscript is needed to improve by adding new references to benefit international readers of the Journals.- it was added new references

Round 2

Reviewer 1 Report

The authors have worked on the manuscripts and addressed the changes requested. Some comments and changes to be looked into include:

1. Line 107, line 112, please consider writing only the first author, followed by et al.

2. Lines 165-167, all genus and species name should be written in italics

3. Fig. 1 map, the text inside is still small. Also indicate north. In the legend

please specify what the coloured lines represent.

4. line 203, avoid using 'we'. 'whence', look for alternatives

5. Lines 209-223, survey questions can be placed in a table format, Did the questions have options, or were they open ended. If they had options, then the survey questions could be attached as a supplementary material.

6. Page 9 , fig 2, specify/acknowledge source of data in the figure legend

7. line 329 avoid using 'we'

8. line 360, change to '950 respondents were interviewed'

9. fig , line 389, check for a more appropriate legend rather than a statement.

10. line 461 to 463- not sure what the authors mean by 'quite closely consistent'. Please rephrase to express idea more clearly.

11. In results when referring to the survey question, specify as Question 1, rather than first question, second question.

Author Response

Dear Reviewer, answers to you comments are marked in yellow!

1.Line 107, line 112, please consider writing only the first author, followed by et al. - changed, marked in yellow, p.3

2. Lines 165-167, all genus and species name should be written in italics-changed, marked in yellow, p.4, 8

3. Fig. 1 map, the text inside is still small. Also indicate north. In the legend please specify what the coloured lines represent. -changed, marked in yellow

4. line 203, avoid using 'we'. 'whence', look for alternatives -changed, marked in yellow, p.6

5. Lines 209-223, survey questions can be placed in a table format. Did the questions have options, or were they open ended. If they had options, then the survey questions could be attached as a supplementary material.- There are no options in answer. Questions are open.

6. Page 9 , fig 2, specify/acknowledge source of data in the figure legend- (Source:Department of Statistics for Akmola region)

 7. line 329 avoid using 'we' -changed, marked in yellow, p.10

8. line 360, change to '950 respondents were interviewed' -changed, p.11

9. fig , line 389, check for a more appropriate legend rather than a statement. -We wouldn’t like to invest to the environment, p.12

10. line 461 to 463- not sure what the authors mean by 'quite closely consistent'. Please rephrase to express idea more clearly. -"match with the results of scientists", p.15 

11. In results when referring to the survey question, specify as Question 1, rather than first question, second question. - changed, p. 11-13

Reviewer 2 Report

The Authors addressed all required comments requested. I have only one concern about English editing.

Author Response

Dear Reviewer, we edited the English version as much as possible.

Reviewer 5 Report

Dear Authors,

please carefully correct the paper. comments are in attached file.

the graph and equations and citations needs to be correct.

Regards,

Author Response

Dear Authors, please carefully correct the paper. comments are in attached file.

the graph and equations and citations needs to be correct.

Regards,

Dear Reviewer, changes to your comments are marked in blue in the text!

Thanks a lot for your comments!

  1. remove the border frame from round of image- removed, p.5
  2. what is this space, distinguish by "," - distinguished by ",", p.5
  3. ?- deleted space, p.6
  4. change "we" - paraphrased, p. 6,10- marked in yellow as the same comments were from the 1st reviewer
  5. is it equation? - yes, it's equation, changed multiply into dot
  6. delete - p.8 note is deleted
  7. signs "*", "x" - signs of multiply changed into dot, p. 7,9
  8. reference - changed , p.16-18

Reviewer 6 Report

The authors, I am grateful for your hard work and did a good job in reviewing the manuscript, thus improving its overall quality. Most of the comments have been addressed. However, there are some minor revisions that need attention before publication:

- Both in the abstract and throughout the manuscript, the authors did not follow my previous recommendation for the overall pattern of the research manuscript considered for publication in International Journals.

The results are not incorporated in the abstract section.

Some parts of the materials and methods are incorporated in the results section.

The results of the present study are not discussed in the discussion section.

Author Response

The authors, I am grateful for your hard work and did a good job in reviewing the manuscript, thus improving its overall quality. Most of the comments have been addressed. However, there are some minor revisions that need attention before publication:

- Both in the abstract and throughout the manuscript, the authors did not follow my previous recommendation for the overall pattern of the research manuscript considered for publication in International Journals.

The results are not incorporated in the abstract section.

Some parts of the materials and methods are incorporated in the results section.

The results of the present study are not discussed in the discussion section.

Dear Reviewer!

Sorry, that we didn't notice your previous comments! Thank you very much for your support.!

Answers to your comments are marked in the article in pink colour:

  1. we have changed the name of the article -Assessment of forest ecosystem services in Burabay National park, Kazakhstan: A case study
  2. we have addeded this sentence as the process of the resuls are described in the chapter "Results" - The real value of the multifunctional value of forests has been revealed.

  3. It 's a forest fund p. 5- The Forest Fund is a natural and economic object of federal property, forest relations, management, use and reproduction of forests, representing the totality of forests, forest and non-forest lands within the boundaries established in accordance with forest and land legislation
  4. At the same page there are corrections (in blue) after numerals instead of space p.5
  5. in the Results "Carbon fixation value" we can't shift the paragraph to materials and methods section as everything will be changed. Before your comments this paragraph was in the chapter materials and methods but one of the review asked to move it to the results. 
  6. signes of multiply changed

Thank you very much for your comments!

Round 3

Reviewer 5 Report

Dear Authors, 

Thanks for improving the manuscript. It is ready for publication.

Regards,